# Syndicated Bandits: A Framework for Auto Tuning Hyper-parameters in Contextual Bandit Algorithms

**Qin Ding**
Department of Statistics
University of California, Davis
qding@ucdavis.edu

**Yue Kang**
Department of Statistics
University of California, Davis
yuekang@ucdavis.edu

**Yi-Wei Liu**
Department of Statistics
University of California, Davis
lywliu@ucdavis.edu

**Thomas C. M. Lee**
Department of Statistics
University of California, Davis
tcmlee@ucdavis.edu

**Cho-Jui Hsieh**
Department of Computer Science
University of California, Los Angeles
chohsieh@cs.ucla.edu

**James Sharpnack**
Amazon[*]
Berkeley, CA
jsharpna@gmail.com

## Abstract

The stochastic contextual bandit problem, which models the trade-off between exploration and exploitation, has many real applications, including recommender systems, online advertising and clinical trials. As many other machine learning algorithms, contextual bandit algorithms often have one or more hyper-parameters. As an example, in most optimal stochastic contextual bandit algorithms, there is an unknown exploration parameter which controls the trade-off between exploration and exploitation. A proper choice of the hyper-parameters is essential for contextual bandit algorithms to perform well. However, it is infeasible to use offline tuning methods to select hyper-parameters in contextual bandit environment since there is no pre-collected dataset and the decisions have to be made in real time. To tackle this problem, we first propose a two-layer bandit structure for auto tuning the exploration parameter and further generalize it to the Syndicated Bandits framework which can learn multiple hyper-parameters dynamically in contextual bandit environment. We derive the regret bounds of our proposed Syndicated Bandits framework and show it can avoid its regret dependent exponentially in the number of hyper-parameters to be tuned. Moreover, it achieves optimal regret bounds under certain scenarios. Syndicated Bandits framework is general enough to handle the tuning tasks in many popular contextual bandit algorithms, such as LinUCB, LinTS, UCB-GLM, etc. Experiments on both synthetic and real datasets validate the effectiveness of our proposed framework.

## 1  Introduction

The stochastic contextual bandit problem models the well-known exploration-exploitation dilemma in a repeated game between a player and an environment. At each round, the player sequentially interacts with the environment by pulling an arm from a pool of $K$ arms, where every arm is associated with a

---

[*]Work done prior to joining Amazon.

36th Conference on Neural Information Processing Systems (NeurIPS 2022).

$d$-dimensional contextual feature vector. Only the stochastic reward corresponding to the pulled arm is revealed to the player. The goal of the player is to maximize the cumulative reward or minimize the cumulative regret. Due to the partial feedback setting, the player has to balance between exploitation — pulling the arm that has the best estimated reward so far — and exploration — exploring whether there are uncertain arms that may be better than the current estimated best arm.

With substantial applications in recommender systems [17], online advertising [20], clinical trials [25], etc., bandit algorithms have been extensively studied during the past few decades. In general, there are two exploration techniques, Upper Confidence Bound (UCB) [6, 17, 18] and Thompson Sampling (TS) [4, 5] algorithms. The UCB algorithm addresses the dilemma optimistically by pulling the arm that has the biggest upper confidence bound. The TS algorithm usually assumes that the relationship between the contextual features and rewards follows a prior model and it uses new observations at each round to estimate the posterior model. The player then pulls the arm that has the best estimated reward based on the posterior model. In general, the contextual bandit problems have regret lower bounded by $O(\sqrt{T})$ [13, 17], where $T$ is the total number of rounds. Both UCB and TS algorithms have been shown to achieve optimal regret bounds in (generalized) linear bandit problems [17, 18], kernelized bandit problems [12] and even in the contextual bandit problem with more complicated models such as neural networks [28].

Despite the popularity of the contextual bandit problem, there are some practical issues that prevent it from being used widely in practice. In both UCB and TS, there are hyper-parameters that are unknown to the player. One of the most important hyper-parameters is the exploration parameter, which controls the trade-off between exploration and exploitation. A good choice of the exploration parameter is essential for the algorithm to perform well and for the theory to hold. Another commonly seen hyper-parameter is the regularization parameter $\lambda$ in ridge regression or generalized linear model, which is used to model the relationship between features and rewards in (generalized) linear bandits. In contextual bandit problems with complex models such as neural network, the recently proposed NeuralUCB [28] algorithm has far more than just two hyper-parameters. NeuralUCB also needs to select the network width, network depth, step size for gradient descent to solve the neural networks and gradient descent steps, etc. Due to the nature of the bandit environment, where the decisions have to be made in real time, it is inherently difficult to tune the hyper-parameters by the traditional offline tuning methods, such as cross validation, since when you have decided to use a parameter in partial datasets and make a decision based on this, the regret incurred by this decision will never be reversible in contextual bandit environment. In many prominent bandit works [17, 14, 28, 10], the experiments are conducted by running a grid search on the possible choices of parameters and only the best result is reported. Although the performance of the best grid search results is of academic interest, it is not possible to do grid search in practice. In some other works [15], exploration parameter is set as a sufficient, theoretically derived, and often unknown value, but this value may be too conservative and may not achieve good performances in practice, which can be seen from the experiments in Table 1.

In this work, we first propose a two-layer bandit structure that can automatically tune the exploration parameter dynamically from the observed data. The two-layer bandit structure has its similarities to the bandit-over-bandit (BOB) algorithm [11] proposed for the non-stationary stochastic bandit problems, where it uses BOB idea to successfully adapt its sliding-window sizes by restarting the algorithms in epochs. Motivated by the two-layer bandit structure we propose in Section 4, we generalize it to the "Syndicated Bandits" framework where there could be multiple hyper-parameters to be tuned in the contextual bandit algorithm. We provide theoretical guarantees for our framework and show that our proposed auto tuning method in general has regret upper bound $\tilde{O}(T^{2/3}) + \tilde{O}(\sum_{l=1}^{L} \sqrt{n_l T})$. Here $L$ is the total number of hyper-parameters to be tuned and $n_l$ is the number of candidates in the tuning set of the $l$-th hyper-parameter. When the unknown theoretical exploration parameter is no bigger than any elements in the tuning set, our proposed framework has optimal regret upper bound $\tilde{O}(\sqrt{T}) + \tilde{O}(\sum_{l=1}^{L} \sqrt{n_l T})$ for UCB-based algorithms. Our framework is general enough to handle tuning tasks in many contextual bandit algorithms as long as the arms to be pulled at round $t$ follows a fixed distribution given the hyper-parameters to be used at this round and the past information. This includes many popular contextual bandit algorithms such as Linear UCB (LinUCB) [17, 1], Linear TS (LinTS) [5, 10], UCB-GLM [18], etc. Our proposed Syndicated Bandits framework is the first work that considers tuning multiple parameters dynamically from observations in the contextual bandit problems with theoretical guarantees. We provide a regret bound that avoids the exponential dependency on the total number of hyper-parameters to be tuned. This is one of the main contributions of our proposed work. In Section 6, we show by experiments that our proposed framework improves

over existing works, as well as the bandit algorithms that use the unknown theoretically derived exploration parameter.

## 2 Related work

There is a rich line of works on multi-armed bandit (MAB) and stochastic contextual bandit algorithms, including (generalized) linear bandits, kernelized bandits and neural bandits, etc. Most of them follow the UCB and TS exploration techniques. We refer the readers to [17, 18, 4, 5, 10, 12, 28] for the seminal works regarding the bandit problems. There are many previous works that utilize algorithms in the stochastic MAB [23] setting to solve the hyper-parameter optimization problem [21, 19]. There are also some online hyper-parameter tuning works such as [24], however, those mainly focuses on reducing the training cost for tuning parameters of neural networks online and they are not considering minimizing the cumulative regret in contextual bandit problems. In the following, we will only pay attention to related works on the tuning tasks in stochastic contextual bandits.

[22] proposed a meta-learning method for learning exploration parameters in contextual bandit problems. It learns a good exploration strategy in synthetic datasets and applies it to the real contextual bandit problems by an imitation study. The meta-learning algorithm is compared with seven baseline contextual bandit algorithms and achieves good empirical results. We note that this algorithm cannot learn the exploration parameters adaptively from observations in the contextual bandit environment. In [9], the authors first proposed OPLINUCB and DOPLINUCB algorithms to learn exploration parameters dynamically. OPLINUCB treats the possible choices of hyper-parameters as arms and uses a standard MAB TS algorithm to choose parameters. It then uses the chosen parameter in the contextual bandit algorithm. However, this method does not have theoretical guarantee in general, since the MAB TS only works when the rewards of the candidate hyper-parameters in the tuning set stay stationary over time. For hyper-parameter selections in contextual bandit problems, the best exploration parameter does not stay the same all the time. This is because in later rounds, when the learning is sophisticated, less exploration is better. However, in the beginning, more exploration is preferred due to the uncertainty. This non-stationary nature in tuning hyper-parameters makes the performance of OPLINUCB unstable in practice. DOPLINUCB is a similar tuning method as OPLINUCB, except that it uses the CTree algorithm to select hyper-parameters at each round. It is shown in [9] that DOPLINUCB does not outperform OPLINUCB in stationary contetxual bandit environments, where the reward-feature model does not change over time.

Another close line of literature is on model selections in bandit algorithms. [16] tackles the feature selection problem in bandit algorithms and achieve $O(T^{2/3}d_*^{1/3})$ where $d_*$ is the total number of optimal features. [3] uses the corralling idea to create a master algorithm to choose the best bandit model from a set of $M$ base models. Hyper-parameter tuning problem can be formulated as a model selection problem in [3], where we can treat bandit algorithms with different hyper-parameters as the base models. The theoretical regret bound of the corralling idea [3] is $O(\sqrt{MT} + MR_{\max})$, where $M$ is the total number of base models and $R_{\max}$ is the maximum regret of $M$ base models if they were to run alone. This means that the regret bound will be exponentially dependent on the total number of hyper-parameters to be tuned. In addition, if there is one hyper-parameter in the tuning set that gives linear regret of the algorithm, then $R_{\max}$ is linear in $T$ which makes the corralling idea have linear regret in worst case. Our algorithm is also much more efficient than the corralling idea when $M$ is big. The corralling idea requires updating all $M$ base models/ algorithms at each round. However, our algorithm only needs to update the selected model/ bandit algorithm with selected hyper-parameter at each round. When the time complexity of updating the model/ algorithm is big, the corralling idea is expensive. For example, if we tune configurations for UCB-GLM, the corralling idea needs $O(MT^2d)$ time, while the time complexity of our algorithm is only $O(MT + T^2d)$.

We address here that none of the previous works can tune multiple parameters dynamically from observations. Although OPLINUCB [9] and the corralling idea [3] can treat all the hyper-parameters as a single parameter and set the tuning set as all the possible combinations of hyper-parameters. This will lead to exponential number of configurations which may not be efficient in both computation and theoretical regret bounds. Our proposed Syndicated framework avoids the exponential regret bound.

**Notations:** For a vector $x \in \mathbb{R}^d$, we use $\|x\|$ to denote its $l_2$ norm and $\|x\|_A := \sqrt{x^T A x}$ for a positive-definite matrix $A \in \mathbb{R}^{d \times d}$. Finally, we denote $[n] := \{1, 2, \ldots, n\}$.

# 3 Preliminaries

We study the hyper-parameter selection tasks in a stochastic contextual bandit problem with $K$ arms, where $K$ can be an infinite number. Assume there are in total $T$ rounds, at each round $t \in [T]$, the player is given $K$ arms, represented by a set of feature vectors $\mathcal{A}_t = \{x_{t,a} | a \in [K]\} \subset \mathbb{R}^d$, $\mathcal{A}_t$ is drawn IID from an unknown distribution with $\|x_{t,a}\| \leq 1$ for all $t \in [T]$ and $a \in [K]$, where $x_{t,a}$ is a $d$-dimensional feature vector that contains the information of arm $a$ at round $t$. The player makes a decision by pulling an arm $a_t \in [K]$ based on $\mathcal{A}_t$ and past observations. We make a common regularity assumption as in [14, 18], i.e. there exists a constant $\sigma_0 > 0$ such that $\lambda_{\min} \left( \mathbb{E}[\frac{1}{k} \sum_{a=1}^{k} x_{t,a} x_{t,a}^\top] \right) > \sigma_0$. The player can only observe the rewards of the pulled arms. Denote $X_t := x_{t,a_t}$ as the feature vector of the pulled arm at round $t$ and $Y_t$ the corresponding reward. We assume the expected rewards and features follow a model $\mathbb{E}[Y_t | X_t] = \mu(X_t^T \theta^*)$, where $\mu(\cdot)$ is a known model function and $\theta^*$ is the true but unknown model parameter. When $\mu(x) = x$, this becomes the well-studied linear bandits problem. When $\mu(\cdot)$ is a generalized linear model or a neural network, this becomes the generalized linear bandits (GLB) and neural bandits respectively.

Without loss of generality, we assume that there exists a positive constant $S$ such that $\|\theta^*\| \leq S$. We also assume the mean rewards $\mu(x_{t,a}^T \theta^*) \in [0, 1]$ and observed rewards $Y_t \in [0, 1]$. This is a non-critical assumption, which can be easily relaxed to any bounded interval. If $\mathcal{F}_t = \sigma(\{a_s, \mathcal{A}_s, Y_s\}_{s=1}^t \cup \mathcal{A}_{t+1})$ is the information up to round $t$, we assume the observed rewards follow a sub-Gaussian distribution with parameter $\sigma^2$, i.e., $Y_t = \mu(X_t^T \theta^*) + \epsilon_t$, where $\epsilon_t$ are independent random noises that satisfy $\mathbb{E}[e^{b\epsilon_t} | \mathcal{F}_{t-1}] \leq \frac{b^2 \sigma^2}{2}$ for all $t$ and $b \in \mathbb{R}$. Denote $a_t^* = \text{argmax}_{a \in [K]} \mu(X_t^T \theta^*)$ as the optimal arm at round $t$ and $x_{t,*}$ as its corresponding feature, the goal is to minimize the cumulative regret over $T$ rounds defined in the following equation.

$$R(T) = \sum_{t=1}^{T} \left[ \mu(x_{t,*}^T \theta^*) - \mu(X_t^T \theta^*) \right]. \tag{1}$$

For linear bandits where $\mu(x) = x$, classic bandit algorithms such as LinUCB [1, 17] and LinTS [2] compute an estimate of the model parameter $\hat{\theta}_t$ using ridge regression with regularization parameter $\lambda > 0$, i.e., $\hat{\theta}_t = V_t^{-1} \sum_{s=1}^{t-1} X_s Y_s$, where $V_t = \lambda I_d + \sum_{s=1}^{t-1} X_s X_s^T$. Shown by [1], with probability at least $1 - \delta$, the true model parameter $\theta^*$ is contained in the following confidence set,

$$\mathcal{C}_t = \left\{ \theta \in \mathbb{R}^d : \|\theta - \hat{\theta}_t\|_{V_t} \leq \alpha(t) \right\}, \tag{2}$$

where

$$\alpha(t) = \sigma \sqrt{d \log \left( \frac{1 + t/\lambda}{\delta} \right)} + S\sqrt{\lambda}. \tag{3}$$

To balance the trade-off between exploration and exploitation, there are in general two techniques. For example, in linear bandits, LinUCB explores optimistically by pulling the arm with the maximum upper confidence bound, while LinTS adds randomization by drawing a sample model from the posterior distribution and pulls an arm based on it.

$$a_t = \underset{a}{\text{argmax}} \, x_{t,a}^T \hat{\theta}_t + \alpha(t) \|x_{t,a}\|_{V_t^{-1}}, \tag{LinUCB}$$

$$\theta_t^{\text{TS}} \sim N(\hat{\theta}_t, \alpha(t)^2 V_t^{-1}) \quad \text{and} \quad a_t = \underset{a}{\text{argmax}} \, x_{t,a}^T \theta_t^{\text{TS}}. \tag{LinTS}$$

In the following, we call $\alpha(t)$ the exploration parameter. As suggested by the theories in [1, 17], a conservative choice of the exploration parameter is to follow Equation 3. However, in Equation 3, the upper bound of the $l_2$ norm of the model parameter $S$ and the sub-Gaussian parameter $\sigma$ are unknown to the player, which makes it difficult to track theoretical choices of the exploration parameter.

In Table 1, we show the cumulative regret of LinUCB [1, 17] and LinTS [5] in a simulation study with $d = 5$, $T = 10000$ and $K = 100$. Rewards are simulated from $N(x_{t,a}^T \theta^*, 0.5)$. The model parameter $\theta^*$ and feature vectors $x_{t,a}$ are drawn from Uniform$(-\frac{1}{\sqrt{d}}, \frac{1}{\sqrt{d}})$. Two scenarios are considered in this table. In the first scenario, the feature of each arm keeps the same over $T$ rounds. While in the second scenario, the features are re-simulated from Uniform$(-\frac{1}{\sqrt{d}}, \frac{1}{\sqrt{d}})$ at different rounds. We

run a grid search of the exploration parameter in $\{0, 0.5, 1, \ldots, 10\}$ and report the best grid search result, as well as the results using the theoretical exploration parameter given by Equation 3 (last column in Table 1). As we shall see in Table 1, the best exploration parameter is not the same for different scenarios. Therefore, which exploration parameter to use is an instance-dependent problem and the best exploration parameter should always be chosen dynamically based on the observations. Meanwhile, theoretical exploration parameters do not always give the best performances from Table 1. On the other hand, in many other works where the model of contextual bandit problem is more complex, such as the generalized linear bandit [14], neural bandit [28], there may be many more hyper-parameters than just $\alpha(t)$.

Table 1: Averaged cumulative regret (cum. reg.) and standard deviation (std) of the cumulative regret based on 5 repeated experiments. "Fixed" means that the feature vectors of each arm are fixed over time. "Changing" means that the features are re-simulated from the same distribution at each round.

| Algorithm | Feature type | Best $\alpha(t)$ | Cum. reg. (std) | Theoretical cum. reg. (std) |
|---|---|---|---|---|
| LinUCB | Fixed | 4.0 | 357.21 (188.72) | 364.99 (151.54) |
| | Changing | 1.5 | 312.69 (42.53) | 582.59 (523.60) |
| LinTS | Fixed | 1.5 | 336.44 (137.01) | 576.83 (110.48) |
| | Changing | 3.5 | 352.79 (109.84) | 488.99 (141.34) |

## 4 A two-layer bandit structure for tuning exploration parameters

In the previous section, we have discussed that the best hyper-parameters should be instant-dependent. In this section, we propose a two-layer bandit structure to automatically learn the best hyper-parameter from data at each round. We will take learning the best exploration parameter as an example. However, we want to emphasize that this structure can also be applied to learn other single hyper-parameter.

We randomly select arms for the first $T_1$ rounds to warm up the algorithm. For all rounds later, in this two-layer bandit structure, the top layer follows an adversarial MAB policy, namely, the EXP3 algorithm [7]. Assume $J$ is the tuning set of all the possible exploration parameters. At each round $t > T_1$, the top layer will select a candidate exploration parameter $\alpha_{i_t} \in J$, where $\alpha_i$ is the $i$-th element in the set $J$ and $i_t$ is the selected index at round $t$. The bottom layer runs the contextual bandit algorithm based on the selected exploration parameter $\alpha_{i_t}$. Details are listed in Algorithm 1.

### 4.1 Regret analysis

Given all the past information $\mathcal{F}_{t-1}$, denote $a_t(\alpha_j | \mathcal{F}_{t-1})$ as the pulled arm when the exploration parameter is $\alpha_j$ at round $t$. Denote $X_t(\alpha_j | \mathcal{F}_{t-1}) = x_{t,a_t(\alpha_j | \mathcal{F}_{t-1})}$ as the corresponding feature vector under $\mathcal{F}_{t-1}$. Note that in our algorithm, $X_t := X_t(\alpha_{i_t} | \mathcal{F}_{t-1})$ when $t > T_1$. To analyze the cumulative regret, we first decompose the regret defined in Equation 1 into three parts:

$$\mathbb{E}[R(T)] = \mathbb{E}\left[\sum_{t=1}^{T}\left(\mu\left(x_{t,*}^T\theta\right) - \mu\left(X_t^T\theta\right)\right)\right] = \underbrace{\mathbb{E}\left[\sum_{t=T_1+1}^{T}\left(\mu\left(x_{t,*}^T\theta\right) - \mu\left(X_t(\alpha^*|\mathcal{F}_{t-1})^T\theta\right)\right)\right]}_{\text{Quantity (A)}}$$

$$+ \underbrace{\mathbb{E}\left[\sum_{t=T_1+1}^{T}\left(\mu\left(X_t(\alpha^*|\mathcal{F}_{t-1})^T\theta\right) - \mu\left(X_t(\alpha_{i_t}|\mathcal{F}_{t-1})^T\theta\right)\right)\right]}_{\text{Quantity (B)}} + \underbrace{\mathbb{E}\left[\sum_{t=1}^{T_1}\left(\mu\left(x_{t,*}^T\theta\right) - \mu\left(X_t^T\theta\right)\right)\right]}_{\text{Quantity (C)}},$$

where $\mu(\cdot)$ is the reward-feature model function and $\alpha^* \in J$ is some arbitrary candidate exploration parameter in $J$. Quantity (A) is the regret of the contextual bandit algorithm that runs with the same hyper-parameter $\alpha^*$ under the past history $\mathcal{F}_{t-1}$ generated from our tuning strategy every round. Quantity (B) is the extra regret paid to tune the hyper-parameter. Quantity (C) is the regret paid for random exploration in warm-up phases and is controlled by the scale of $O(T_1)$. We show by Lemma 1 and Theorem 1 below that our auto tuning method in Algorithm 1 does not cost too much in selecting parameters in most scenarios under mild conditions.

---

**Algorithm 1** A Two-layer Auto Tuning Algorithm

---

**Input**: time horizon $T$, warm-up length $T_1$, candidate hyper-parameter set $J = \{\alpha_i\}_{i=1}^n$.

1: Randomly choose $a_t \in [K]$ and record $X_t, Y_t$ for $t \in [T_1]$.
2: Initialize exponential weights $w_j(T_1 + 1) = 1$ for $j = 1, \ldots, n$.
3: Initialize the exploration parameter for EXP3 as $\beta = \min\left\{1, \sqrt{\frac{n \log n}{(e-1)T}}\right\}$.
4: **for** $t = (T_1 + 1)$ **to** $T$ **do**
5:     Update probability distribution for pulling candidates in $J$

$$p_j(t) = \frac{\beta}{n} + (1 - \beta) \frac{w_j(t)}{\sum_{i=1}^n w_i(t)}$$

6:     $i_t \leftarrow j \in [n]$ with probability $p_j(t)$.
7:     Run the contextual bandit algorithm with hyper-parameter $\alpha(t) = \alpha_{i_t}$ to pull an arm. For example, pull arms according to the following equations

$$a_t = \underset{a=1,\ldots,K}{\operatorname{argmax}} \, x_{t,a}^T \hat{\theta}_t + \alpha_{i_t} \|x_{t,a}\|_{V_t^{-1}} \qquad \text{(LinUCB)}$$

$$\theta_t^{\text{TS}} \sim N(\hat{\theta}_t, \alpha_{i_t}^2 V_t^{-1}) \quad \text{and} \quad a_t = \underset{a}{\operatorname{argmax}} \, x_{t,a}^T \theta_t^{\text{TS}}. \qquad \text{(LinTS)}$$

8:     Observe reward $Y_t$ and update the components in the contextual bandit algorithm.
9:     Update EXP3 components: $\hat{y}_t(j) \leftarrow 0$ if $j \neq i_t$, $\hat{y}_t(j) \leftarrow Y_t/p_j(t)$ if $j = i_t$, and

$$w_j(t + 1) = w_j(t) \times \exp\left(\frac{\beta}{n} \hat{y}_t(j)\right).$$

10: **end for**

---

Since the arms pulled by the contextual bandit layer also affect the update of the EXP3 layer in Algorithm 1, the result of EXP3 algorithm is not directly applicable to bounding Quantity (B). We modify the proof techniques in [7] and present the proof details in Appendix.

**Lemma 1.** *Assume given the past information $\mathcal{F}_{t-1}$ and the hyper-parameter to be used by the contextual bandit algorithm at round $t$, the arm to be pulled follows a fixed distribution. For a random sequence of hyper-parameters $\{\alpha_{i_1}, \ldots, \alpha_{i_T}\}$ selected by the EXP3 layer in Algorithm 1, and arm $a_t(\alpha_{i_t})$ is pulled in the contextual bandit layer at round $t$, we have*

$$\max_{\alpha \in J} \mathbb{E}\left[\sum_{t=1}^T \mu\left(X_t(\alpha | \mathcal{F}_{t-1})^T \theta\right)\right] - \mathbb{E}\left[\sum_{t=1}^T \mu\left(X_t(\alpha_{i_t} | \mathcal{F}_{t-1})^T \theta\right)\right] \leq 2\sqrt{(e-1)nT \log n},$$

*where $J = \{\alpha_1, \ldots, \alpha_n\}$ is the tuning set of the hyper-parameter and $|J| = n$.*

To bound Quantity (A), we note that we are not able to use any existing regret bound in the literature directly since the past information $\mathcal{F}_{t-1}$ here is based on the sequence of arms pulled by our auto-tuning algorithm instead of the arms generated by using $\alpha^*$ at each round, and the history would affect the update of bandit algorithms. We overcome this challenge by noticing that the consistency of $\hat{\theta}_t$ plays a vital role in most of the proofs for (generalized) linear bandits, and this consistency could hold after a warm-up period or with large exploration rate. Therefore, we can expect a tight bound of the cumulative regret by using the same exploration parameter even under another line of observations $\mathcal{F}_{t-1}$ with sufficient exploration. Another crux of proof is that the regret is usually related to $\|x_t\|_{V_t^{-1}}$, which can be similarly bounded after sufficient exploration. After we bound Quantity (A), combing Lemma 1, we get the following theorem.

**Theorem 1.** *Assume given the past information $\mathcal{F}_{t-1}$ generated from our proposed algorithm for arm selection and the hyper-parameter to be used by the contextual bandits, the arm to be pulled follows a fixed distribution. For UCB and TS based generalized linear bandit algorithms with exploration hyper-parameters (LinUCB, UCB-GLM, LinTS, ect.), the regret bound of Algorithm 1 satisfies*

*(1) $\mathbb{E}[R(T)] = \tilde{O}(T^{2/3}) + O(\sqrt{n(T - T_1) \log n})$ given the warm-up length $T_1 = \tilde{O}(T^{2/3})$.*

(2) *For UCB-based bandits, if the theoretical exploration parameter $\alpha(T)$ is no larger than any element in $J$, then it holds that $\mathbb{E}[R(T)] = \tilde{O}(\sqrt{T}) + O(\sqrt{nT \log n})$ with $T_1 = 0$.*

(3) *If $\mathcal{A}_t$ is a convex set, and the smallest principal curvature in any neighborhood of the optimal vector $x_{t,*} \in \mathcal{A}_t$ on $\mathcal{A}_t$ can be lower bounded by some positive constant $c$, then $\mathbb{E}[R(T)] = \tilde{O}(T^{4/7}) + O(\sqrt{n(T-T_1)\log n})$ after a warming-up period of length $T_1 = O(T^{4/7})$.*

**Remark 1.** *We could expect a similar result for TS-based bandit algorithms as in Theorem 1 (2), and we offer an intuitive explanation in Appendix 4. Moreover, the conditions in Proposition 1 (3) could be easily verified in many cases. For example, it holds when $\mathcal{A}_t = \{x \in \mathbb{R}^d : \|x\| \leq a\}, \forall a > 0$.*

## 5   The Syndicated Bandits framework for selecting multiple hyper-parameters

There can be multiple hyper-parameters in the contextual bandit algorithm. For example, in linear bandit algorithms such as LinUCB[1, 17] and LinTS [5], exploration parameter $\alpha$ and the regularization parameter of the ridge regression $\lambda$ are both hyper-parameters to be tuned. In more recent contextual bandit works, there could be even more than two hyper-parameters. For example, NeuralUCB algorithm [28], which is proposed for the contextual bandit problems with a deep neural network model, has many tuning parameters such as the network width, network depth, step size for gradient descent, number of steps for gradient descent, as well as exploration parameter and regularization parameter $\lambda$, etc. Another example can be found in [14], where an efficient SGD-TS algorithm is proposed for generalized linear bandits, the number of tuning parameters is also more than two.

A naive strategy to auto-tune multiple hyper-parameters is to use Algorithm 1 and let the tuning set $J$ contain all the possible combinations of the hyper-parameters. Assume there are in total $L$ hyper-parameters $\alpha^{(1)}, \alpha^{(2)}, \ldots, \alpha^{(L)}$. For all $l \in [L]$, if the tuning set for $\alpha^{(l)}$ is defined as $J_l = \{\alpha_1^{(l)}, \ldots, \alpha_{n_l}^{(l)}\}$, where $n_l$ is the size of the corresponding tuning set. Then there are in total $\Pi_{l=1}^{L} n_l$ possible combinations. Based on Lemma 1, the

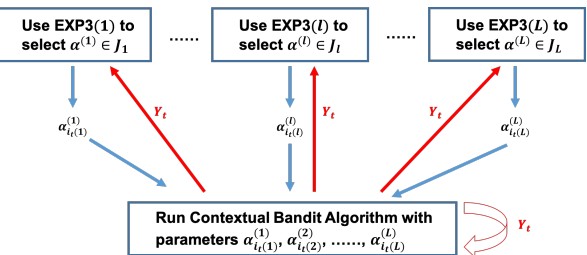

Figure 1: Illustration of the Syndicated Bandits.

extra regret paid to tune the hyper-parameters (Quantity (B)) is upper bounded by $\tilde{O}(\sqrt{\Pi_{l=1}^{L} n_l T})$. Therefore, the aforementioned naive approach makes the regret increase exponentially with the number of tuning parameters. To mitigate this issue, we propose the Syndicated Bandits framework that can deal with multiple hyper-parameters while avoiding the exponential dependency on the number of tuning parameters in regret bounds.

We create $L + 1$ bandit instances in this framework. In the bottom layer, the contextual bandit algorithm is used to decide which arm to pull. On top of the contextual bandit layer, there are $L$ EXP3 bandits, denoted as EXP3($l$) for $l \in [L]$. Each EXP3 algorithm is responsible for tuning one hyper-parameter only. At round $t$, if $i_t(l)$ is the index of the hyper-parameters in $J_l$ selected by the EXP3($l$) bandit and the selected hyper-parameter is denoted as $\alpha_{i_t(l)}^{(l)}$ for $l \in [L]$, then the contextual bandit algorithm in the bottom layer will use these parameters to make a decision and receive a reward based on the pulled arm. The reward is fed to all the $L + 1$ bandits to update the components. Illustration of the algorithm and more details are presented in Figure 1 and Algorithm 2 in Appendix.

### 5.1   Regret analysis

At round $t$, given all the past information $\mathcal{F}_{t-1}$, denote $a_t(\alpha_{j_1}^{(1)}, \ldots, \alpha_{j_L}^{(L)} | \mathcal{F}_{t-1})$ as the arm pulled by the contextual bandit algorithm if the parameters are chosen as $\alpha^{(l)} = \alpha_{j_l}^{(l)}$ for all $l \in [L]$ and let $X_t(\alpha_{j_1}^{(1)}, \ldots, \alpha_{j_L}^{(L)} | \mathcal{F}_{t-1})$ be the corresponding feature vector. Recall that $\mu(\cdot)$ is the reward-feature

model function, then for an arbitrary combination of hyper-parameters $(\alpha_*^{(1)}, \ldots, \alpha_*^{(L)})$,

$$
\mathbb{E}[R(T)] = \sum_{t=1}^{T_1} \mathbb{E}\left[\mu\left(x_{t,*}^T\theta\right) - \mu\left(X_t^T\theta\right)\right] + \sum_{t=T_1+1}^{T} \mathbb{E}\left[\mu\left(x_{t,*}^T\theta\right) - \mu\left(X_t(\alpha_*^{(1)}, \ldots, \alpha_*^{(L)}|\mathcal{F}_{t-1})^T\theta\right)\right]
$$

$$
+ \sum_{t=T_1}^{T} \mathbb{E}\left[\mu\left(X_t(\alpha_*^{(1)}, \ldots, \alpha_*^{(L)}|\mathcal{F}_{t-1})^T\theta\right) - \mu\left(X_t(\alpha_{i_t(1)}^{(1)}, \alpha_*^{(2)}, \ldots, \alpha_*^{(L)}|\mathcal{F}_{t-1})^T\theta\right)\right]
$$

$$
+ \sum_{t=T_1}^{T} \mathbb{E}\left[\mu\left(X_t(\alpha_{i_t(1)}^{(1)}, \alpha_*^{(2)}, \ldots, \alpha_*^{(L)}|\mathcal{F}_{t-1})^T\theta\right) - \mu\left(X_t(\alpha_{i_t(1)}^{(1)}, \alpha_{i_t(2)}^{(2)}, \alpha_*^{(3)} \ldots |\mathcal{F}_{t-1})^T\theta\right)\right]
$$

$$
+ \cdots + \sum_{t=T_1}^{T} \mathbb{E}\left[\mu\left(X_t(\alpha_{i_t(1)}^{(1)}, \ldots, \alpha_{i_t(L-1)}^{(L-1)}, \alpha_*^{(L)}|\mathcal{F}_{t-1})^T\theta\right) - \mu\left(X_t(\alpha_{i_t(1)}^{(1)}, \ldots, \alpha_{i_t(L)}^{(L)}|\mathcal{F}_{t-1})^T\theta\right)\right].
$$

The first quantity represents the regret from pure exploration. The second quantity in the above decomposition is the regret of the contextual bandit algorithm that runs with the same hyper-parameters $\alpha_*^{(1)}, \ldots, \alpha_*^{(L)}$ under the past history $\mathcal{F}_{t-1}$ generated from our tuning strategy every round. The next $L$ quantities in the decomposition are the regret from tuning parameters in the EXP3 layer, which can be bounded using similar techniques in Lemma 1. However, the correlations between parameters are more complicated in the analysis now. Formally, we provide the following Theorem to guarantee the performance of the Syndicated Bandits framework. Proofs are deferred to the Appendix.

**Theorem 2.** *Assume given the past information $\mathcal{F}_{t-1}$ and the hyper-parameters to be used by the contextual bandit algorithm at round $t$, the arm to be pulled by the contextual bandit algorithm follows a fixed distribution. Then the auto tuning method in Algorithm 2 with warm-up length $T_1 = O(T^{2/3})$ has the following regret bound in general:*

$$
\mathbb{E}[R(T)] \leq \tilde{O}(T^{2/3}) + O\left(\sum_{l=1}^{L} \sqrt{n_l(T - T_1)\log n_l}\right).
$$

**Remark 2.** *Note this result avoids the exponential dependency on the number of hyper-parameters to be tuned in regret. When the hyper-parameters to be tuned are the exploration parameter $\alpha$ and the regularization parameter $\lambda$ of the (generalized) linear model, we also have the same conclusions as in Theorem 1 (3). Please refer to Appendix A.3 for a formal statement and its proof.*

**Remark 3.** *Without any assumptions, Algorithm 2 has its regret dependent on $d$ as $O(d^3 + dT^{2/3})$ for both UCB and TS. In practice, usually $d << T$.*

## 6 Experimental results

We show by experiments that our proposed methods outperform various contextual bandit algorithm using the theoretical exploration parameter, as well as existing tuning methods. We compare different hyper-parameter selection methods in three popular contextual bandit algorithms, LinUCB [1, 17], LinTS [5] and UCB-GLM [18] with a logistic model. In practice, we set the warm-up length as $T_1 = 0$ and tune both exploration parameters and regularization parameters. We compare the following hyper-parameter selection methods. **Theoretical-Explore [1]**: At round $t$, this method uses the true theoretical exploration parameter $\alpha(t)$ defined in Equation 3; **OP [9]**: We make simple modifications of OPLINUCB to make it applicable to tune exploration parameters for LinUCB, LinTS and UCB-GLM; **Corral [3]**: This method uses the corralling idea to tune the exploration parameter only. **Corral-Combined [3]**: This method treats bandits with different combinations of the exploration parameter and regularization parameter $\lambda$ as base model and uses the corralling idea to tune the configurations; **TL (Our work, Algorithm 1)**: This is our proposed Algorithm 1, where we use the two-layer bandit structure to tune the exploration parameter only; **TL-Combined (Our work, Algorithm 1)**: This method tunes both the exploration parameter $\alpha$ and the regularization parameter $\lambda$ using Algorithm 1, but with the tuning set containing all the possible combinations of $\alpha$ and $\lambda$; **Syndicated (Our work, Algorithm 2)**: This method keeps two separate tuning sets for $\alpha$ and $\lambda$ respectively. It uses the Syndicated Bandits in Algorithm 2.

We set the tuning set for exploration parameter $\alpha$ as $\{0, 0.01, 0.1, 1, 10\}$ and set the tuning set for regularization parameter $\lambda$ as $\{0.01, 0.1, 1\}$ in TL-Combined, Corral-Combined and Syndicated.

**Algorithm 2** The Syndicated Bandits Framework for Auto Tuning Multiple Hyper-parameters

---

**Input**: time horizon $T$, warm up length $T_1$, candidate hyper-parameter set $\{J_l\}_{l=1}^{L}$.

1: Randomly choose $a_t \in [K]$ and record $X_t, Y_t$ for $t \in [T_1]$.

2: Initialize exponential weights $w_j^{(l)}(T_1 + 1) = 1$ for $t = 1$, $j = 1, \ldots, n_l$ and $l = 1, \ldots, L$.

3: Initialize the parameters for all EXP3 layers as $\beta_l = \min\left\{1, \sqrt{\frac{n_l \log n_l}{(e-1)T}}\right\}$.

4: **for** $t = (T_1 + 1)$ **to** $T$ **do**

5:     Update probability distribution for pulling candidates in $J_l$

$$p_j^{(l)}(t) = \frac{\beta_l}{n_l} + (1 - \beta_l) \frac{w_j^{(l)}(t)}{\sum_{i=1}^{n_l} w_i^{(l)}(t)}$$

6:     $i_t(l) \leftarrow j \in [n_l]$ with probability $p_j^{(l)}(t)$ for all $l = 1, \ldots, L$.

7:     Run the contextual bandit algorithm with hyper-parameters $\alpha^{(l)} = \alpha_{i_t(l)}^{(l)}$ to pull an arm.

8:     Observe reward $Y_t$ and update the components in contextual bandit algorithms.

9:     Update all $L$ EXP3 bandits: $\hat{y}_t^{(l)}(j) \leftarrow 0$ if $j \neq i_t(l)$. Otherwise, $\hat{y}_t^{(l)}(j) \leftarrow Y_t / p_j^{(l)}(t)$.

10:     For all $l = 1, \ldots, L$, let $w_j^{(l)}(t + 1) = w_j^{(l)}(t) \times \exp\left(\frac{\beta_l}{n_l} \hat{y}_t^{(l)}(j)\right)$.

11: **end for**

---

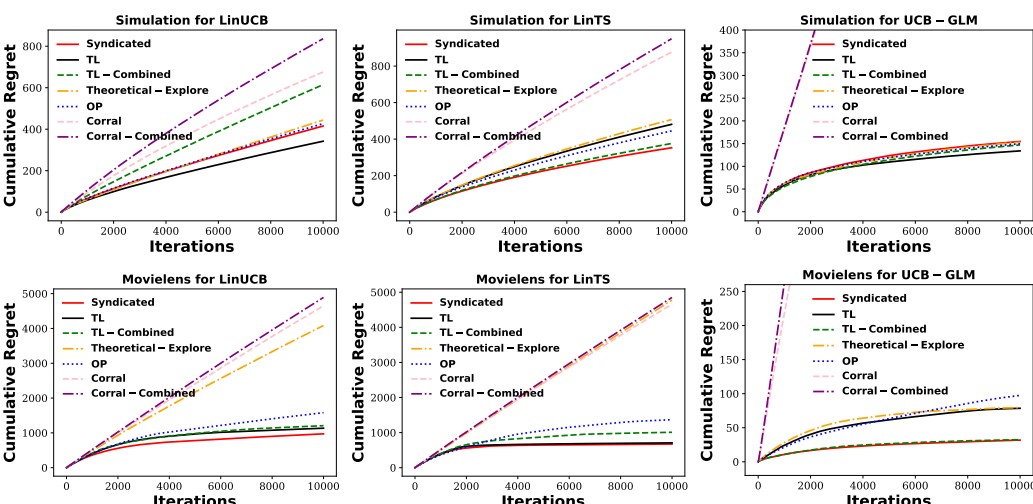

Figure 2: Comparison of hyper-parameters selection methods in LinUCB, LinTS and UCB-GLM.

For Theoretical-Explore, OP and TL, since they only tune the exploration parameter, we set the regularization parameter as $\lambda = 1$. In all the experiments below, the total number of rounds is $T = 10,000$. We run the comparisons on both simulations and the benchmark Movielens 100K real datasets. Due to limited space, the descriptions of the dataset settings are deferred to Appendix A.4. Averaged results over 10 independently repeated experiments are reported below.

From Figure 2, we observe: 1) When tuning only one hyper-parameter (exploration parameter in our experiments), the proposed method outperforms previous tuning methods. Further, the theoretical exploration parameter does not perform well and it tends to be too conservative in practice, which is consistent with the results we show in Table 1. 2) When tuning multiple hyper-parameters, previous methods do not apply. We found using the Syndicated Bandits framework usually outperforms TL-Combined and is significantly better than Corral-Combined method which has exponential regret with respect to number of tuning parameters. 3) Using Syndicated Bandits to tune multiple hyper-parameters usually outperforms tuning one parameter only. This demonstrates a practical need of auto-tuning multiple hyper-parameters in bandit algorithms. See Appendix for additional experiments on the tuning 3 hyper-parameters in SGD-TS [14].

# 7 Conclusion

In this paper, we propose a two-layer bandit structure for auto tuning the exploration parameter in contextual bandit algorithms, where the offline tuning is impossible. To further accommodate multiple hyper-parameters tuning tasks in contextual bandit algorithms with complicated models, we generalize our method to the Syndicated Bandits framework. This is the first framework that can auto-tune multiple hyper-parameters dynamically from observations in contextual bandit environment with theoretical regrets that avoids exponential dependency on the total number of hyper-parameters to be tuned. We show that our proposed algorithm can obtain $\tilde{O}(T^{2/3})$ regret in general and has optimal $\tilde{O}(\sqrt{T})$ regret for UCB-based algorithms when the all candidates in the tuning set is greater than the theoretical exploration parameter. Our work is general enough to handle the tuning tasks in many contextual bandit algorithms. Experimental results also validate the effectiveness of our proposed work.

## Acknowledgments and Disclosure of Funding

We are grateful for the insightful comments from the anonymous reviewers and area chair. This work was partially supported by the National Science Foundation under grants CCF-1934568, DMS-1811405, DMS-1811661, DMS-1916125, DMS-2113605, DMS-2210388, IIS-2008173 and IIS-2048280. CJH is also supported by Samsung, Google, Sony and the Okawa Foundation.

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
