# OpenReview forum: "Syndicated Bandits: A Framework for Auto Tuning Hyper-parameters in Contextual Bandit Algorithms"
_NeurIPS.cc/2022/Conference — NeurIPS 2022 Accept_

### Official Review · Reviewer_6M9s · 2022-07-08

**Rating:** 7
**Confidence:** 4
**Soundness:** 3 good
**Presentation:** 3 good
**Contribution:** 3 good

**Summary:**

This paper addressed an important and practical problem for bandit algorithms: how to auto-tune the exploration parameter to automate exploration-exploitation tradeoff to release the full power of bandit algorithms. The authors contributes knowledge on bandit hyperparameter auto-tuning with a bandit-over-bandit framework, applicable to important bandit algorithms class including LinUCB and LinTs. The proposed framework shares same flavor of EXP3 algorithms, in the sense that the superiority of a candidate exploration parameter is captured by a quantity as weight and then competes with other candidates. The validity of presented approach is backed by regret analysis and experiments.

**Questions:**


I have one question came from my personal interest. It would be of my great appreciation if the authors share their insights.

My question came from picking "one" class of bandit algorithms from a pool of candidate bandit algorithms". You may already share same experience that, in your Algorithm 1 and 2, the " time horizon T and warm up length T1" are typically subject to change in practice. We may pick T and T1 subjectively, and I think we don't have a good strategy to find a pair of "great (T , T1)" to pick the "best bandit algorithms from candidate pool".

Your framework invites to think about the possibility of "bandit-ovet-bandit-over-bandit" flavor framework to do automatic decision making. It would be great if you can share your thoughts on this point, if you also feel this is an interesting problem.

**Limitations:**


The only interesting limitation came to my mind is the problem "bandit algorithm candidate auto selection" I elaborated in the questions section. This is not count as a limitation of current submission.

**Strengths And Weaknesses:**

Strengths:

The writing is clear and easy to follow.
The authors did great job on presenting evidences of their claims, and the message is clear conveyed.


Weaknesses:

I am satisfied with the submitted version.

---

> ### Author Response · Authors · 2022-08-02
> **Thanks for your insightful comments. Please see below for the responses to your questions.**
>
> In our experiments, $T_1$ is set to 0 in practice and we can see it already achieves good performances empirically. $T_1$ is introduced here to analyze the theoretical regret bounds. Bandit-over-bandit-over-bandit is definitely an awesome idea and that was actually the initial idea of this paper. Technically, we can introduce more layers of bandits to pick the best hyper-parameters in the next lower layer. However, that also introduces one more hyper-parameter in the top layer. How to introduce more layers of bandits and make the algorithm not rely on any hyper-parameters in the top layer is definitely an important and interesting future work.

---

### Official Review · Reviewer_nYJA · 2022-07-11

**Rating:** 6
**Confidence:** 4
**Soundness:** 3 good
**Presentation:** 3 good
**Contribution:** 4 excellent

**Summary:**

This paper proposed a two-layer bandits framework for automatic tuning of multiple hyper-parameters in contextual bandits algorithms. The bottom layer hosts the original bandits algorithm (UCB or TS-based), and the top layer is a set of EXP3 bandits algorithm, each of which is responsible for tuning one of the hyper-parameter in the parameter set. By sending the same global reward signals from the bottom layer to each of the top-layer bandits algorithm, the paper shows that the resulting regret bound has a linear (and not exponential) dependency on the number of hyper-parameters to be tuned. Experiments on synthetic and real-world data are performed for benchmarking and showing the advantage of the proposed framework in terms of cumulative regrets.

**Questions:**

- The syndicated bandits share similarity to a multi-agent setting, where each agent performs its dedicated task but the agents share the same global reward signals so as to act cooperatively. I wonder if you could draw any connection here to explain why it works much better in regret bound than using a single agent to tune the multiple parameters.

- In the experiments, there are only two hyper-parameters with 15 possible combinations. Is the advantage seen in the plot from Syndicated  over TL-combined is due to the limited number of iterations? It would be better and more convincing to demonstrate the advantage of Syndicated if there were a test case with more hyper-parameters.

**Limitations:**

N.A.

**Strengths And Weaknesses:**

Strengths
- This paper can be of significant practical interest to a broad audience group who are interested in and using bandits algorithms. The auto-tuning algorithm for single and multiple hyper-parameters with a desirable regret bound provides a very useful tool for successful deployment of bandits algorithms.

- The proposed algorithm comes with a regret bound analysis that offers assurance that its learning efficiency is much better than naively treating the multiple hyper-parameters as a single parameter with more options.

Weaknesses
- Besides theoretical analysis of the regret bound, it is also important to provide intuition on why the change from using a single top-layer bandit to multiple parallel bandits can allow us to avoid the exponential dependency of regret on the number of hyper-parameters.

- The experiments were all conducted on test cases with only two parameters. It would be great to have concrete examples of more hyper-parameters and associated empirical results.

---

> ### Author Response · Authors · 2022-08-02
> **Thanks for your insightful comments. Please see below for the responses to your questions.**
>
> 1. Thanks for mentioning the multi-agent setting. Multiple bandits in parallel allow them to simultaneously learn the best parameter together based on the same rewards received from a taken action. So one observation allows us to learn the best candidate for each hyper-parameter in parallel. However, if we only use one bandit to learn the best combination of hyper-parameters, then this needs information on all possible combinations, which is inefficient in utilizing the information collected from the bottom layer. We will add more explanations in the final revision.
>
>
> 2. We do have simulation results in Appendix A.5 that tunes a contextual bandit algorithm SGD-TS [14] with more hyper-parameters. This algorithm has 3 hyper-parameters including 2 exploration parameters and 1 step size for the SGD optimizer. The total number of combinations of candidate hyper-parameters is 100. In the experiments, Syndicated bandits and the two-layer algorithms proposed in our paper still perform the best.

---

### Official Review · Reviewer_UTTP · 2022-07-12

**Rating:** 6
**Confidence:** 4
**Soundness:** 3 good
**Presentation:** 3 good
**Contribution:** 2 fair

**Summary:**

This work studies the problem of hyperparameter tuning for contextual bandit algorithms.  Due to the unique properties of contextual bandits (no pre-collected dataset, and the decision have to be made online), offline tuning methods can not be directly applied. The authors proposed a two-layer framework to automatically tune the exploration hyperparameter in the process of bandit learning. A upper regret bound of $\tilde O(T^{2/3})$ is proved for the proposed method in general. A $\tilde O(T^{1/2})$ is derived for the UCB-based algorithm when the theoretical exploration parameter is no bigger than any elements in the tuning set.

**Questions:**

- Why not perform the tuning for NeurUCB?
- What are the  requirements of the candidate hyperparameter set, both theoretically and theoretically

**Limitations:**

See weakness.

**Strengths And Weaknesses:**

Strength:
- The hyperparameter tuning problem is quite important and the proposed method is reasonable.
- The theoretical analysis is sound to me.

Weakness:

- The empirical evaluation is quite toy.  (1) Why not also tuning more advanced bandit models, e.g., NeurUCB?  (2) In addition to the Movielens dataset, performing the evaluation on the Yahoo news recommendation dataset will be more convincing.
- The   $\tilde O(T^{2/3})$ upper regret bound is quite weak.
- About the underlying assumptions on the search space of hyperparameter configurations is not clearly discussed. For example, the candidate hyperparameter set must be finite?  If it must be finite, how large it can be? Can it be larger than T? In the experiments, the size of the set is pretty small. It would be helpful and interesting if the author can add more discussions on more on the requirements of the candidate hyperparameter set, both theoretically and theoretically.

---

> ### Author Response · Authors · 2022-08-02
> **Thanks for your insightful comments. Please see below for the responses to your questions.**
>
> 1. (1). We do have simulation results in Appendix A.5 that tune a more advanced algorithm SGD-TS [14]. This algorithm has 3 hyper-parameters including 2 exploration parameters and 1 step size for the SGD optimizer. The total number of combinations of candidate hyper-parameters is 100. In the experiments, Syndicated bandits and the two-layer algorithms proposed in our paper still perform the best.
>
> (2). For experiments on the Yahoo dataset, see the table below for the averaged cumulative rewards (the higher the better). The preprocessing of the Yahoo dataset is the same as in [14] and the total number of rounds is T=2881 (since the data is collected from Yahoo from May01 to May10 2009 with 2881 timestamps). The three numbers in the column “rewards” are the results of tuning LinUCB, LinTS and UCB-GLM respectively. From the table below, we can see that our proposed Syndicated bandits and Two-layer bandits (TL) algorithms perform well on the Yahoo dataset. The Syndicated bandits algorithm is the best for LinUCB. TL is only slightly worse than OP [9] for LinTS where OP does not have theoretical guarantees. TL is only slightly worse than Theoretical methods for UCB-GLM, however, theoretical parameters are unknown to players in practice.
>
>
>
>     ----------------------------------—--------------------------------------------------
>      Methods                      Rewards (LinUCB; LinTS; UCB-GLM)
>     ----------------------------------—--------------------------------------------------
>     Theoretical                     220.58;   214.50;  222.78
>     TL                              217.20;   216.89;  219.88
>     OP                              217.69;   217.62;  218.54
>     Syndicated                      220.73;   214.79;  216.21
>     TL-Combined                     218.21;   214.51;  216.08
>     Corral                          218.83;   216.60;  215.49
>     Corral-combined                 217.80;   214.62;  213.30
>     ----------------------------------—---------------------------------------------------
>
>
>
> 2. The $\tilde O(T^{2/3})$ regret bounds are for general cases and it doesn’t have any assumptions on the tuning set. To clarify the contribution of this regret analysis, we can compare it with the corralling idea [3], which has regret $ O( \sqrt {MT} + MR_{\max}) $. Here $M$ is the total number of base models and $R_{\max}$ is the regret of the worst model in the tuning set. So the regret of the corralling idea is $\tilde O(\sqrt T)$ if only the worst model in the tuning set has regret $\tilde O(\sqrt T) $. Our algorithm can achieve $\tilde O(\sqrt T)$ regret as well for single-hyperparameter problems if we have similar conditions as the corralling idea. See conclusion (2) in Theorem 1 (line 231, page 6 of the paper). This is because when the theoretical exploration parameter is no larger than any elements in the tuning set $J$, then the worst model in the tuning set has $\tilde O(\sqrt T)$ regret upper bounds. In general cases, where we don’t have assumptions on the tuning set, we have $\tilde O(T^{2/3})$ regret bounds but the corralling idea can get regrets as bad as $O(T)$ if the worst model in the tuning set is bad.
>
> 3. We do not have assumptions for the tuning set in experiments. We do have simulation results in appendix A.5 with a bigger tuning set. This algorithm has 3 hyper-parameters including 2 exploration parameters and 1 step size for the SGD optimizer. The total number of hyper-parameter combinations is 100. But the theoretical regret does depend on the tuning set:
>
> (1). Without any assumptions on the tuning set (See Theorem 2), the regret bound is $ \tilde O(T^{2/3}) + O( \sum_l \sqrt {n_l T \log n_l})$.  Here $n_l$ is the number of candidates for the $l$-th hyper-parameter in the tuning set. Based on this result, with a bigger tuning set, we get bigger theoretical regret, but also a higher chance to include the optimal configurations in the tuning set. How to choose the tuning set in practice is a trade-off between the above two considerations.
>
> (2). In conclusion (2) of Theorem 1, we discussed that when the tuning set has all the elements greater than the theoretical exploration parameter, the regret is $ \tilde O(\sqrt T) + O( \sqrt {n T \log n})$. This is a special case where we can achieve $\tilde O(\sqrt T)$ regrets. But similar to before, the 2nd quantity in the regret bound also requests us to trade-off between higher theoretical regret and higher chance of including the optimal parameters in the tuning set.

---

### Official Review · Reviewer_vjwY · 2022-07-13

**Rating:** 6
**Confidence:** 3
**Soundness:** 3 good
**Presentation:** 4 excellent
**Contribution:** 3 good

**Summary:**

This paper studied hyperparameter tuning for contextual bandits. The authors proposed a two-layer bandit structure to tune the exploration parameter where the top layer applies EXP3 to select a hyperparameter as adversarial MAB and the bottom layer is the contextual bandits to be tuned. The authors then generalized it to Syndicated Bandits to tune multiple hyperparameters. The proposed method achieve $O(T^{2/3}+\sum_l\sqrt{n_l T})$ regret for tuning UCB and TS based algorithms. Empirical evaluations show that proposed method outperforms baselines.


**Questions:**

Besides the questions in weakness, I have two clarification questions:

1. Is expect regret upper bound for tuning TS based algorithm defined as Bayesian regret or frequentist regret? I tried to follow the proof of Theorem 1 and it looks like frequentist regret to me, but the proof mixed UCB and TS and is not very clear.

2. What is the dependency of regret on feature dimension $d$? Do they the same as bottom layer algorithm, e.g., linear for UCB and $d^{1.5}$ for TS?

**Limitations:**

See weakness.


**Strengths And Weaknesses:**

Strength:

Hyperparameter tuning for contextual bandits is an important and interesting problem. The idea of using adversarial bandits to select best hyperparameter is simple yet very intuitive. The proposed method has a non-trivial (although may not be optimal) regret guarantee and could be also of interest to practice. The paper is also well written and the main paper is easy to follow.

Weakness:

I have two concerns regarding the paper:

1. The regret is only analyzed for UCB and TS based generalized linear bandits and is not general. The proof technique heavily relies on analyzing confidence ellipsoid of generalized linear bandits. Is it possible to extend the two-layer method for tuning other specific algorithms such as elimination based algorithm or adversarial bandits such as EXP3 or even general bandit algorithm, just like other model selection baselines?


2. The $O(T^{2/3})$ regret seems not optimal. The key challenge and novelty in the proof is bounding Quantity (A) where the history depends on the sequences of pulled hyperparameters instead of contextual bandits using a consistent hyperparameter. However, corralling idea can achieve $O(\sqrt{T})$ regret as discussed in related work and is strictly better if only tuning one hyperparameter. Does the author suspect this $O(T^{2/3})$ is tight and inevitable in general case, if not allowing exponential dependency on the number of hyperparameters?

3. A minor suggestion is the proof is a bit hard to follow as it mixed UCB and TS and I would suggest the authors make a better presentation.

---------------After rebuttal---------------
Most of my questions are answered and I increased my score accordingly.  I suggest the authors clearly discuss the regret dependency on dimension $d$ in the final version.

---

> ### Author Response · Authors · 2022-08-02
> **Thanks for your insightful comments. Please see below for the responses to your questions.**
>
> Regarding your comments about weaknesses:
>
> 1. Our algorithm is applicable to elimination based algorithms or adversarial bandits such as EXP3 empirically. However, the theoretical analysis of these will require additional work and is an interesting future work. In this work, we mainly focus on contextual bandit algorithms.
>
> 2. The regret of corralling idea is $ O( \sqrt {MT} + MR_{\max}) $, where $M$ is the total number of base models and $R_{\max}$ is the regret of the worst model in the tuning set. So the regret of the corralling idea is $\tilde O(\sqrt T)$ only when the worst model in the tuning set has regret $\tilde O(\sqrt T)$. Our algorithm can achieve $\tilde O(\sqrt T)$ regret as well for single-hyperparameter problems if we have similar conditions as the corralling idea. See conclusion (2) in Theorem 1 (line 231, page 6 of the paper). This is because when the theoretical exploration parameter is no larger than any element in the tuning set $J$, the worst model in the tuning set has $\tilde O(\sqrt T)$ regret upper bounds. In general cases, when we don’t have assumptions on the tuning set, we have $\tilde O(T^{2/3})$ regret bounds but the corralling idea can get regrets as bad as $O(T)$ if the worst model in the tuning set is bad.
>
> 3. We revised the proofs in our paper to make them more clear. See “rebuttal revision” for more details. We will continue to polish this and make the proofs more clear in the final revision.
>
> Regarding your questions:
>
> 1. It’s the frequentist regret. Our theory establishes the expected regret and it’s not dependent on observed information.
>
> 2. Without any assumptions, the regret dependency on $d$ is $\tilde O(d^3 + d T^{2/3})$ for both UCB and TS. In practice, usually $d << T$.

---

> > ### Comment · Reviewer_vjwY · 2022-08-09
> > **Thank you for the response**
> >
> > I thank authors for the answering my questions and revising the paper. I am increasing my score accordingly. I suggest the authors clearly discuss the regret dependency on dimension $d$ in the final version since dependency on dimension is important for analyzing contextual bandits.

---

> > > ### Author Response · Authors · 2022-08-09
> > > **Thanks for your suggestions!**
> > >
> > > Thank you for your insightful suggestions! We will definitely add discussions about the regret dependency on $d$ in the final revision.

---

### Meta-Review · Area_Chair_v3Tf · 2022-08-23

**Recommendation:** Accept
**Confidence:** Certain

**Metareview:**

Thank you for submitting your paper to NeurIPS! This paper studies the important practical challenge of parameter tuning in real-world contextual bandit implementations. The authors propose a bandit-over-bandit framework: the bottom layer is the original bandit algorithm (e.g. LinUCB or LinTs), and the top layer is an adversarial EXP3 algorithm for choosing the best hyperparameter. The resulting regret bound has a linear (and not exponential) dependence on the number of hyper-parameters. There was consensus that the algorithm is intuitive and novel, and the experiments support its promise. I am pleased to recommend acceptance.

**Award:**

No

---

### Decision · Program_Chairs · 2022-09-14

Accept